# Calculating age-specific prevalence rates of female genital mutilation / cutting (FGM/C) for use as an input variable in extrapolation calculations and as predictors of future prevalence in countries of origin

**Sean Callaghan** *

School of Criminology, University of Leicester, Leicester, United Kingdom

* smc85@le.ac.uk

## Abstract

This paper proposes a refined method for calculating age-specific prevalence rates of Female Genital Mutilation/Cutting (FGM/C) to enhance the accuracy of estimates calculated using Yoder and Van Baelen's Extrapolation-of-FGM/C-Countries-Prevalence-Data method. Previous studies, particularly in the United States, have faced limitations, including the failure to disaggregate prevalence data by age and overlooking historical trends. To address these limitations, this study outlines a comprehensive seven-step approach. Using Ethiopia as a case study, prevalence rates were calculated and aligned with target migrant population data. This involved adjusting age cohorts, extrapolating prevalence to younger age groups, and considering historical trends. Results demonstrate significant differences compared to previous estimates, indicating overestimation of girls at risk of FGM/C in some studies. The proposed method offers a standardized approach applicable beyond the United States, potentially improving estimates globally. By providing nuanced prevalence data, this method contributes to better understanding the true prevalence of FGM/C in migrant populations. This same method can also be used to predict future trends in FGM/C and other practices.

## 1. Introduction

Female Genital Mutilation/Cutting (FGM/C) is defined by the World Health Organization as the 'partial or total removal of external female genitalia or other injury to the female genital organs for non-medical reasons' [1]. Globally, at least 230 million women and girls are estimated to be cut [2]. There is documented evidence of the practice in over 90 countries–a third of which (mostly in Africa) base prevalence data on nationally representative surveys and a further third of which base prevalence data on indirect estimates of FGM/C within the migrant population resident in the country [3]. The global female migrant population is estimated to be 135 million [4, 5] with analysis suggesting that more than 7.3 million of those migrant

**Data Availability Statement:** All relevant data are within the paper and its Supporting Information files.

**Funding:** The author(s) received no specific funding for this work.

**Competing interests:** The authors have declared that no competing interests exist.

women and girls are impacted by FGM/C [6]. It is further estimated that more than four hundred thousand migrant girls and women are impacted by FGM/C in the United States [7], while in Europe it is estimated that more than six hundred thousand migrant women have undergone FGM/C and a further 189,438 girls are at risk [8].

The most widely used process for estimating the scale and distribution of the FGM/C-impacted population in migrant populations is Yoder and Van Baelen's 'Extrapolation of FGM/C Countries' Prevalence Data' method [9]. While there is a clear refinement of the method evident in the literature, at its core the extrapolation method relies on three input variables: the prevalence in the country of origin (*Pr*), the absolute number of migrants as enumerated in the national census form the target population in the country under investigation (*TP*), and an estimation of the impact of migration and acculturation on prevalence based on qualitative studies with immigrants and often expressed as a set of scenarios (*AI*). The basic 'Extrapolation of FGM/C Countries' Prevalence Data' formula calculates the potentially impacted population as follows:

$$Impacted\ population = \sum_{n=1}^{x}(Pr_n \times TP_n \times (1 - AI))$$

Where:

$Pr_n$ = the FGM/C Prevalence in a specific country of origin (*n*)

$TP_n$ = the target population in the country of residence associated with a specific country of origin (*n*)

*AI* = a composite variable indicating the acculturation impact associated with migration in a range from 0 (no impact) to 1 (total impact)

x = the number of countries of origin under investigation

For example, in its simplest form, and assuming no impact of migration and acculturation on prevalence (*AI* = 0), this formula can be applied to an imaginary target population (*TP*) comprising 1,000 Nigerian and 1,000 Somali females. Since prevalence (*Pr*) is estimated at 15.1% and 99.2% respectively [10, 11] the number of impacted females would be 1,143 (1,000 x 15.1% x 1 + 1,000 x 99.2% x 1).

It is the prevalence variable (*Pr*) in that equation that is the focus of analysis in this paper.

## 1.1 Sources of prevalence data in countries of origin

The data most often used as the basis for the prevalence variable (*Pr*) is extracted from either Multiple Indicator Cluster Surveys ('MICS') or Demographic and Health Surveys ('DHS'), both of which provide nationally representative household surveys covering several health and wellbeing indictors specific to women aged 15 to 49. The USAID-funded DHS Program was established in the mid-1980s, and the UNICEF-supported MICS programme was established a decade later. Between them they cover 138 countries, providing the most comprehensive global health-related dataset currently available. The FGM/C modules used by the MICS and DHS are very similar, MICS asking 24 questions and DHS asking 21 questions. Questions include knowledge of and attitudes towards the practice as well as specifics–age, cutter and type–of each respondent's own FGM/C status and that of her children, thus making survey results largely comparable across time, country and implementing agency. In addition to these MICS and DHS surveys, this study added several other nationally representative surveys that were independently commissioned by national statistics agencies. Each of these independent surveys were, however, based on the same 'gold standard' methodologies developed for the MICS and DHS surveys [12]. While sample sizes varied across the surveys included in this study from 3,040 in Cote d'Ivoire in DHS 1998–99 to 30,660 in Iraq in MICS 2018, all of the

surveys referenced in this analysis selected samples that were nationally representative at a 95 percent level of confidence [13] with error margins on the FGM/C data calculated at between 0.2 and 1.78 percent based on the number of women interviewed and the number of women in the population at the time.

Since inclusion of the FGM/C module in DHS and MICS surveys is voluntary, not all countries collect FGM/C data while others only have a single survey documenting FGM/C prevalence. As a result, prevalence data from 27 countries with at least two surveys that included the FGM/C module were used in this analysis. In total 120 nationally representative surveys spanning a period of 28 years from 1994 to 2022, were included. Age-specific FGM/C prevalence data from 74 surveys compiled by DHS were downloaded from STATcompiler [14]. Age-specific FGM/C prevalence data for a further 43 surveys were extracted from data tables in individual MICS reports [15]. The remaining three age-specific FGM/C prevalence datasets were extracted from national statistics agency reports for Egypt [16], Eritrea [17] and Somalia [10].

These surveys are not without their challenges and limitations. Surveys are generally conducted on a five-year rolling basis, and while MICS and DHS provide standardised FGM/C modules, these are optional–countries may (or may not) choose to include them, resulting in some gaps in the data. Furthermore, household surveys, by definition, exclude those members of society who do not live in households, thus resulting in some skewing of the results. More recently, it has become apparent that the direct-questioning method used by both MICS and DHS is resulting in some level of underreporting of more sensitive data–including FGM/C–due to increasing social-desirability bias [18–20], especially in contexts where FGM/C is illegal [21]. Since these population surveys in countries of origin provide a critical input variable to the extrapolation method, this analysis seeks to mitigate this bias.

## 1.2 Prevalence values used in previous estimates of FGM/C

Reviewing the prevalence data used in previous studies based on the extrapolation method highlights two significant shortcomings. The first and most fundamental is the failure to disaggregate the prevalence data by age. In the United States ('US'), three studies applied the average national country-of-origin prevalence calculated by MICS or DHS to the whole of the impacted migrant population [22–24] in their extrapolation calculations, while Goldberg et al. [25], used the national average for those 20 years of age or older and the 15–19-year-old prevalence value from the same MICS or DHS survey for those below the age of 20 for their extrapolation,. In so doing, each of the authors assumed a relatively stable rate of cutting over time, yet in the context of falling prevalence, as will be shown in this analysis, this methodological choice resulted in an underestimation of the women living with FGM/C and an overestimation of the children at risk of FGM/C. In contrast to the national prevalence used in the US studies, European researchers consistently use prevalence data disaggregated by five-year age cohorts, from 15 to 49 years of age, as presented by the DHS and MICS for their extrapolation calculations. Furthermore, in line with Yoder et al. [26], most European studies use the prevalence in the 45–49-year-old cohort in the latest DHS or MICS survey for those over the age of 50, unlike the US studies, which use the MICS or DHS calculated national average. Similarly to Goldberg et al. [25], most European researchers apply the latest 15–19-year-old-cohort prevalence from the relevant DHS or MICS survey to those below 15 to predict potential risk. Ortensi et al. [27] take a more nuanced approach, introducing the idea of recalculating the latest DHS and MICS prevalence data to align the five-year age-cohorts with the age structure of the target-population data under investigation. This approach was further refined by Ortensi and Menonna [28], who extrapolated the recalculated prevalence trends to age groups below 15 years.

The second limitation lies in the fact that the prevalence values used in previous extrapolation studies in both Europe and the US were based solely on the latest DHS and MICS data available at the time of the study. This gives rise to two potential errors: the first is assuming a stable prevalence, thus missing historical trends in the prevalence data and falling into the same trap as one using the national average, while the second opens the analysis to potential underreporting errors due to the increasing social-desirability bias evident in more recent surveys.

Addressing these limitations calls for prevalence values (*Pr*) that are disaggregated by age, take historical and future trends into account, and align the calculated prevalence values with the target migrant-population data (*TP*).

## 2. Method: Calculating age-specific prevalence

This paper proposes refinements to the current method by which the prevalence variable (*Pr*) is calculated in order to more comprehensively address the potential underestimation of the number of women living with FGM/C and overestimation of the number of children at risk of FGM/C evident in current extrapolation calculations. The refined method proposed in this paper consists of seven steps:

Step 1: Identify available 15–49-year-old prevalence data.
Step 2: Calculate a temporary 10–14-year-old prevalence value.
Step 3: Calculate the offset to align the prevalence and population data.
Step 4: Calculate age-specific, aligned prevalence data.
Step 5: Extrapolate prevalence down to ages 0–4.
Step 6: Extrapolate prevalence to the older age groups.
Step 7: Calculate the age-specific prevalence means.

To illustrate the proposed method, this paper outlines detailed, step-by-step calculations for estimating FGM/C prevalence in the Ethiopian migrant population in 2019. This demonstrated solution is then applied to other countries with nationally representative surveys in the results section.

### 2.1 Step 1: Identify available 15–49-year-old prevalence data

At the time of this analysis, there were three nationally representative surveys available reporting FGM/C prevalence in Ethiopia. These were published by the DHS. Each survey includes age-disaggregated prevalence data for women aged 15–49 as shown in Table 1 below.

### 2.2 Step 2: Calculate a temporary 10–14-year-old prevalence value

Temporary 10–14-year-old prevalence values ($^{Temp}Pr_{[10\ to\ 14]}$) were calculated for each survey for use in the offset calculations in Step 3 of the process. The resultant values, shown in Table 2, were based on the following calculation:

$$^{Temp}Pr_{[10\ to\ 14]} = {}^{Original}Pr_{[15\ to\ 19]}/{}^{Original}Pr_{[20\ to\ 24]} \ X \ ^{Original}Pr_{[15\ to\ 19]}$$

**Table 1.  Age-disaggregated prevalence data extracted from Ethiopian DHS surveys.**

| Ethiopian Survey | Prevalence Age 15–19 (%) | Prevalence Age 20–24 (%) | Prevalence Age 25–29 (%) | Prevalence Age 30–34 (%) | Prevalence Age 35–39 (%) | Prevalence Age 40–44 (%) | Prevalence Age 45–49 (%) |
|---|---|---|---|---|---|---|---|
| DHS 2000 | 70.7 | 78.3 | 81.4 | 86.1 | 83.6 | 85.8 | 86.8 |
| DHS 2005 | 62.1 | 73.0 | 77.6 | 78.0 | 81.2 | 81.6 | 80.8 |
| DHS 2016 | 47.1 | 58.6 | 67.6 | 76.9 | 74.6 | 72.6 | 78.7 |

**Table 2. Age-disaggregated prevalence data with temporary 10–14-year-old prevalence.**

| Ethiopian Survey | *Temp Age 10–14 (%)* | Age 15–19 (%) | Age 20–24 (%) | Age 25–29 (%) | Age 30–34 (%) | Age 35–39 (%) | Age 40–44 (%) | Age 45–49 (%) |
|---|---|---|---|---|---|---|---|---|
| DHS 2000 | 63.7 | 70.7 | 78.3 | 81.4 | 86.1 | 83.6 | 85.8 | 86.8 |
| DHS 2005 | 52.8 | 62.1 | 73.0 | 77.6 | 78.0 | 81.2 | 81.6 | 80.8 |
| DHS 2016 | 37.9 | 47.1 | 58.6 | 67.6 | 76.9 | 74.6 | 72.6 | 78.7 |

## 2.3 Step 3: Calculate the offset to align the prevalence and population data

Since the Ethiopia DHS surveys used in our example were conducted several years apart, the data needs to be manipulated so the age cohorts align. For example, while the 15–19-year-old cohort from the DHS in 2000 corresponds to the 20–24-year-old cohort from the DHS in 2005, aligning the 2016 data requires a more complex calculation. Furthermore, the target population (*TP*) for this illustrative example was enumerated in 2019 thus necessitating that the prevalence data (*Pr*) be aligned to the *Target Year* 2019. To calculate the offset required to align the datasets with each other and with the target migrant-population data, a quotient (q) indicating the number of quinquennia from the survey year to the target year and remainder (r), expressed as the number of single years, were calculated as follows:

$$d = [\text{Target Year}] - [\text{Survey year}]$$

$$d/5 = q \text{ rem } r$$

Where:

Target Year = the year the target migrant-population data were gathered

q = quotient (the number of quinquennia from the survey year to the target year)

r = remainder (the number of single years)

For example, the DHS 2000 survey took place 19 years prior to the target migrant-population survey in 2019. Therefore, 19 divided by 5 results in three quinquennia (q = 3) with a remainder of four (r = 4). The resultant quotient (q) and remainder (r) for the three input datasets are shown in Table 3 below.

## 2.4 Step 4: Calculate age-specific prevalence data aligned to the target year

Using the quotient (q) and the remainder (r), newly aligned prevalence data (see Table 4) were calculated to align the datasets with each other and with the target migrant-population data. The formula below calculates the prevalence for a specific age group in the target year based on data originally collected by MICS or DHS taking into account that respondents are older in the target year than they were in the year of the survey. In so doing we align each of the age cohorts with each other as they would be in the target year as follows:

$$^{\text{New}}Pr_{[x + (q \times 5>) \text{ to } y + (q \times 5)]} = {}^{\text{Original}}Pr_{[x \text{ to } y]}/5 \text{ x } (5 - r) + {}^{\text{Original}}Pr_{x\text{-}5 \text{ to } y\text{-}5]}/5 \times r$$

## 2.5 Step 5: Extrapolate prevalence down to ages 0–4

It was assumed that the downward trend in prevalence continued, and estimates for each new five-year cohort were extrapolated, mirroring the method of Ortensi and Menonna [28]. This

**Table 3. Resultant quotient and remainder for each survey.**

| Ethiopian Survey | Survey Year | Target Year | *Quotient (q)* | *Remainder (r)* |
|---|---|---|---|---|
| DHS 2000 | 2000 | 2019 | 3 | 4 |
| DHS 2005 | 2005 | 2019 | 2 | 4 |
| DHS 2016 | 2016 | 2019 | 0 | 3 |

**Table 4. Age-specific prevalence data aligned to the target population (2019).**

| Ethiopian Survey | Age 15–19 (%) | Age 20–24 (%) | Age 25–29 (%) | Age 30–34 (%) | Age 35–39 (%) | Age 40–44 (%) | Age 45–49 (%) | Age 50–54 (%) | Age 55–59 (%) | Age 60–64 (%) |
|---|---|---|---|---|---|---|---|---|---|---|
| DHS 2000 | | | | 65.21 | 72.22 | 78.92 | 82.34 | 85.60 | 84.04 | 86.00 |
| DHS 2005 | | | 54.68 | 64.28 | 73.92 | 77.68 | 78.64 | 81.28 | 81.44 | |
| DHS 2016 | 41.55 | 51.70 | 62.20 | 71.32 | 75.98 | 73.80 | 75.04 | | | |

Where [x to y] is [15 to 19]; [20 to 24]. . . [45 to 49]

**Table 5. Extrapolated prevalence values down to ages 15–19 aligned to the target population (2019).**

| Ethiopian Survey | Age 15–19 (%) | Age 20–24 (%) | Age 25–29 (%) | Age 30–34 (%) | Age 35–39 (%) | Age 40–44 (%) | Age 45–49 (%) | Age 50–54 (%) | Age 55–59 (%) | Age 60–64 (%) |
|---|---|---|---|---|---|---|---|---|---|---|
| DHS 2000 | 48.01 | 53.17 | 58.88 | 65.21 | 72.22 | 78.92 | 82.34 | 85.60 | 84.04 | 86.00 |
| DHS 2005 | 39.57 | 46.52 | 54.68 | 64.28 | 73.92 | 77.68 | 78.64 | 81.28 | 81.44 | |
| DHS 2016 | 41.55 | 51.70 | 62.20 | 71.32 | 75.98 | 73.80 | 75.04 | | | |

**Table 6. Extrapolated prevalence values down to ages 0–14 aligned to the target population (2019).**

| Ethiopian Survey | Age 0–4 (%) | Age 5–9 (%) | Age 10–14 (%) | Age 15–19 (%) | Age 20–24 (%) | Age 25–29 (%) | Age 30–34 (%) | . . . | Age 55–59 (%) | Age 60–64 (%) |
|---|---|---|---|---|---|---|---|---|---|---|
| DHS 2000 | 35.34 | 39.14 | 43.35 | 48.01 | 53.17 | 58.88 | 65.21 | . . . | 84.04 | 86.00 |
| DHS 2005 | 24.36 | 28.64 | 33.66 | 39.57 | 46.52 | 54.68 | 64.28 | . . . | 81.44 | |
| DHS 2016 | 21.58 | 26.84 | 33.40 | 41.55 | 51.70 | 62.20 | 71.32 | . . . | | |

**Table 7. Extrapolated prevalence values up to ages 80+ aligned to the target population (2019).**

| Ethiopian Survey | Age 30–34 (%) | . . . | Age 45–49 (%) | Age 50–54 (%) | Age 55–59 (%) | Age 60–64 (%) | Age 65–69 (%) | Age 70–74 (%) | Age 75–79 (%) | Age 80+ (%) |
|---|---|---|---|---|---|---|---|---|---|---|
| DHS 2000 | 65.21 | . . . | 82.34 | 85.60 | 84.04 | 86.00 | 86.80 | 86.80 | 86.80 | 86.80 |
| DHS 2005 | 64.28 | . . . | 78.64 | 81.28 | 81.44 | 80.80 | 80.80 | 80.80 | 80.80 | 80.80 |
| DHS 2016 | 71.32 | . . . | 75.04 | 78.70 | 78.70 | 78.70 | 78.70 | 78.70 | 78.70 | 78.70 |

resulted in extrapolated prevalence values down to ages 15–19 in 2019, as shown in Table 5, based on the following calculation:

$$^{\text{New}}\text{Pr}_{[x \text{ to } y]} = {}^{\text{New}}\text{Pr}_{[x+5 \text{ to } y+5]} / {}^{\text{New}}\text{Pr}_{[x+10 \text{ to } y+10]} \times {}^{\text{New}}\text{Pr}_{[x+5 \text{ to } y+5]}$$

This extrapolation was then continued for the three age cohorts below 15 years of age to estimate the risk of FGM/C in younger girls, as shown in Table 6.

## 2.6 Step 6: Extrapolate prevalence to the older age groups

Applying the method of Yoder *et al.* [26], the original DHS prevalence for the 45–49-year-old cohort was assigned to older cohorts for which there were no calculated prevalence data. This completed the prevalence dataset across the entire target population age range, as shown in Table 7.

## 2.7 Step 7: Calculate the age-specific prevalence means

The realignment of age cohorts highlights inconsistencies in the prevalence data, which potentially distort extrapolated estimates, especially when based on a single survey. Take, for

**Table 8. Inconsistencies in aligned prevalence data once aligned to the target population (2019).**

| Ethiopian Survey | Age 20–24 (%) | Age 25–29 (%) | Age 30–34 (%) | Age 35–39 (%) | Age 40–44 (%) | Age 45–49 (%) | Age 50–54 (%) | Age 55–59 (%) | Age 60–64 (%) | Age 65–69 (%) |
|---|---|---|---|---|---|---|---|---|---|---|
| DHS 2000 | 53.17 | 58.88 | 65.21 | 72.22 | 78.92 | 82.34 | 85.60 | 84.04 | 86.00 | 86.80 |
| DHS 2005 | 46.52 | 54.68 | 64.28 | 73.92 | 77.68 | 78.64 | 81.28 | 81.44 | 80.80 | 80.80 |
| DHS 2016 | 51.70 | 62.20 | 71.32 | 75.98 | 73.80 | 75.04 | 78.70 | 78.70 | 78.70 | 78.70 |

example, the 45–49-year-old data once each of the surveys has been realigned with the 2019 target population data: the data show a 7.3% difference in prevalence across the three surveys, with prevalence in the DHS 2016 survey significantly lower than the corresponding data from 16 years earlier Table 8.

Neither margin of error nor sampling bias can fully account for these inconsistencies. Based on a confidence level of 95%, the margin of error for the FGM/C prevalence variable in each of the surveys was calculated to be 0.79% (DHS2000), 0.83% (DHS2005), and 1.11% (DHS2016). These margins of error do not account for the swings observed in the data. Likewise, according to the DHS, each survey is fully representative and data is adjusted to ensure that sampling bias is minimised [29]. It therefore seems likely that the drop in reported prevalence is at least in part due to increased social-desirability bias. Since later surveys are more likely to underreport prevalence [20], extrapolation calculations based solely on the latest prevalence survey are susceptible to error. To mitigate these inconsistencies in the prevalence data, the mean can be calculated under the assumption of fully representativeness of the underlying survey data for each age cohort, as shown in Tables 9 and 10.

# 3. Results

Using the method outlined above, age-specific prevalence data were calculated for 27 countries based on 120 nationally representative surveys (see Table 11 and S1 Data). The resultant mean age-specific prevalence data that could be applied to 2024 target-population data in an 'Extrapolation of FGM/C Countries' Prevalence Data' method calculation are shown in the Table 12 (appended).

**Table 9. Age-specific prevalence means (ages 0–4 to 40–44) aligned to the target population (2019).**

| Ethiopian Survey | Age 0–4 (%) | Age 5–9 (%) | Age 10–14 (%) | Age 15–19 (%) | Age 20–24 (%) | Age 25–29 (%) | Age 30–34 (%) | Age 35–39 (%) | Age 40–44 (%) |
|---|---|---|---|---|---|---|---|---|---|
| DHS 2000 | 35.34 | 39.14 | 43.35 | 48.01 | 53.17 | 58.88 | 65.21 | 72.22 | 78.92 |
| DHS 2005 | 24.36 | 28.64 | 33.66 | 39.57 | 46.52 | 54.68 | 64.28 | 73.92 | 77.68 |
| DHS 2016 | 21.58 | 26.84 | 33.40 | 41.55 | 51.70 | 62.20 | 71.32 | 75.98 | 73.80 |
| *MEAN* | *27.09* | *31.54* | *36.80* | *43.04* | *50.46* | *58.59* | *66.94* | *74.04* | *76.80* |

**Table 10. Age-specific prevalence means (ages 45–49 to 80+) aligned to the target population (2019).**

| Ethiopian Survey | Age 45–49 (%) | Age 50–54 (%) | Age 55–59 (%) | Age 60–64 (%) | Age 65–69 (%) | Age 70–74 (%) | Age 75–79 (%) | Age 80+ (%) |
|---|---|---|---|---|---|---|---|---|
| DHS 2000 | 82.34 | 85.60 | 84.04 | 86.00 | 86.80 | 86.80 | 86.80 | 86.80 |
| DHS 2005 | 78.64 | 81.28 | 81.44 | 80.80 | 80.80 | 80.80 | 80.80 | 80.80 |
| DHS 2016 | 75.04 | 78.70 | 78.70 | 78.70 | 78.70 | 78.70 | 78.70 | 78.70 |
| *MEAN* | *78.67* | *81.86* | *81.39* | *81.83* | *82.10* | *82.10* | *82.10* | *82.10* |

**Table 11. Nationally representative surveys included in the analysis.**

| Country | National Surveys from which prevalence data were extracted |
| --- | --- |
| Benin | DHS 2001, DHS 2006, DHS 2011–12, MICS 2014 |
| Burkina Faso | DHS 1998–99, DHS 2003, MICS 2006, DHS 2010, DHS 2021 |
| Central African Republic | DHS 1994–95, MICS 2000, MICS 2006, MICS 2010, MICS 2018–19 |
| Chad | DHS 2004, MICS 2010, MICS 2014–15, MICS 2019 |
| Côte d'Ivoire | DHS 1998–99, DHS 2005, MICS 2006, DHS 2011–12, MICS 2016 |
| Egypt | DHS 1995, DHS 2000, DHS 2003, DHS 2005, DHS 2008, DHS 2014, EHIS 2015 |
| Eritrea | DHS 1995, DHS 2002, EPHS 2010 |
| Ethiopia | DHS 2000, DHS 2005, DHS 2016 |
| Gambia | MICS 2005–06, MICS 2010, DHS 2013, MICS 2018, DHS 2019–20 |
| Ghana | DHS 2003, MICS 2006, MICS 2011, MICS 2017–18 |
| Guinea | DHS 1999, DHS 2005, DHS 2012, MICS 2016, DHS 2018 |
| Guinea Bissau | MICS 2006, MICS 2010, MICS 2014, MICS 2018–19 |
| Iraq | MICS 2011, MICS 2018 |
| Kenya | DHS 1998, DHS 2003, DHS 2008–09, DHS 2014, DHS 2022 |
| Liberia | DHS 2013, DHS 2019–20 |
| Mali | DHS 1995–96, DHS 2001, DHS 2006, MICS 2009–10, DHS 2012–13, MICS 2015, DHS 2018 |
| Mauritania | DHS 2000–01, MICS 2007, MICS 2011, MICS 2015, DHS 2019–21 |
| Niger | DHS 1998, DHS 2006, DHS 2012 |
| Nigeria | DHS 2003, MICS 2007, DHS 2008, MICS 2011, DHS 2013, MICS 2016–17, DHS 2018, MICS 2021 |
| Senegal | DHS 2005, DHS 2010–11, DHS 2014, DHS 2015, DHS 2016, DHS 2017, DHS 2018, DHS 2019 |
| Sierra Leone | MICS 2005, DHS 2008, MICS 2010, DHS 2013, MICS 2017, DHS 2019 |
| Somalia | MICS 2006, MICS 2011, SHDS 2020 |
| Sudan | DHS 1989–90, MICS 2010, MICS 2014 |
| Tanzania | DHS 1996, DHS 2004–05, DHS 2010, DHS 2015–16, DHS 2022 |
| Togo | MICS 2006, MICS 2010, DHS 2013–14, MICS 2017 |
| Uganda | DHS 2006, DHS 2011, DHS 2016 |
| Yemen | DHS 1997, DHS 2013 |

## 4. Discussion

The results indicate the likely lifetime risk of FGM/C in each age group at the target date. For those cohorts over the age of cutting this prevalence indicates the estimated proportion of women already cut, while for younger cohorts the prevalence indicates future risk of FGM/C and is expressed as the proportion of children who will eventually be cut.

The data shows a consistent fall in prevalence or risk across age groups in all but three countries–Gambia, Niger and Uganda–where the risk of FGM/C in the 0-4-year-old cohort is higher than the FGM/C prevalence in the over-80-year-old cohort. This negative trend is evident in the underlying MICS and DHS survey data for those countries and is suggestive of either an actual increase in prevalence or margins of error in the source data.

### 4.1 A more accurate input variable

The refinements presented in this paper for calculating the age-specific prevalence of FGM/C for use in estimates based on Yoder and Van Baelen's 'Extrapolation of FGM/C Countries' Prevalence Data' method provide a standardised approach that could be applied in any country-of-residence study.

**Table 12. Mean age-specific FGM/C prevalence for 2024 sorted by prevalence in the 0–4 age group.**

| Country of Origin | Age 0–4 (%) | Age 5–9 (%) | Age 10–14 (%) | Age 15–19 (%) | Age 20–24 (%) | Age 25–29 (%) | Age 30–34 (%) | Age 35–39 (%) | Age 40–44 (%) | Age 45–49 (%) | Age 50–54 (%) | Age 55–59 (%) | Age 60–64 (%) | Age 65–69 (%) | Age 70–74 (%) | Age 75–79 (%) | Age 80 + (%) |
|---|---|---|---|---|---|---|---|---|---|---|---|---|---|---|---|---|---|
| Somalia | 93.18 | 93.86 | 94.54 | 95.24 | 95.94 | 96.73 | 97.77 | 98.36 | 98.52 | 98.81 | 98.89 | 98.66 | 98.66 | 98.90 | 98.90 | 98.90 | 98.90 |
| Egypt | 88.37 | 88.87 | 89.52 | 90.33 | 91.33 | 92.57 | 93.89 | 94.96 | 95.75 | 96.23 | 96.42 | 96.41 | 96.38 | 96.67 | 96.87 | 96.83 | 96.83 |
| Mali | 84.45 | 84.97 | 85.51 | 86.05 | 86.60 | 87.16 | 88.01 | 88.37 | 88.15 | 89.02 | 89.48 | 89.47 | 89.42 | 89.22 | 89.13 | 88.97 | 88.97 |
| Gambia | 80.62 | 79.82 | 79.06 | 78.33 | 77.64 | 77.14 | 76.56 | 75.51 | 75.12 | 75.12 | 75.18 | 75.68 | 75.82 | 75.40 | 75.40 | 75.40 | 75.40 |
| Guinea | 79.86 | 81.92 | 84.04 | 86.24 | 88.52 | 90.89 | 92.89 | 94.77 | 96.45 | 97.58 | 98.39 | 98.66 | 98.70 | 98.96 | 99.00 | 99.00 | 99.00 |
| Sudan | 65.01 | 67.60 | 70.29 | 73.09 | 76.01 | 79.05 | 82.21 | 84.71 | 86.36 | 86.58 | 89.50 | 90.31 | 89.91 | 90.15 | 89.97 | 90.09 | 90.60 |
| Mauritania | 52.27 | 54.06 | 55.97 | 58.01 | 60.20 | 62.66 | 64.88 | 66.44 | 68.39 | 71.44 | 73.66 | 74.53 | 74.58 | 74.53 | 73.58 | 73.58 | 73.58 |
| Eritrea | 41.92 | 45.73 | 49.97 | 54.70 | 59.99 | 65.92 | 72.55 | 79.59 | 85.67 | 89.88 | 92.59 | 93.66 | 94.16 | 94.84 | 94.75 | 95.07 | 95.07 |
| Guinea Bissau | 37.08 | 38.68 | 40.46 | 42.43 | 44.62 | 47.05 | 47.97 | 48.27 | 48.96 | 46.06 | 48.71 | 49.32 | 47.99 | 46.65 | 46.65 | 46.65 | 46.65 |
| Sierra Leone | 24.56 | 30.00 | 36.77 | 45.20 | 55.75 | 69.00 | 80.14 | 89.00 | 94.10 | 95.44 | 96.25 | 96.34 | 96.41 | 96.40 | 96.40 | 96.40 | 96.40 |
| Ethiopia | 23.32 | 27.09 | 31.54 | 36.80 | 43.04 | 50.46 | 58.59 | 66.94 | 74.04 | 76.80 | 78.67 | 81.86 | 81.39 | 81.83 | 82.10 | 82.10 | 82.10 |
| Chad | 22.72 | 24.79 | 27.12 | 29.77 | 32.77 | 36.18 | 39.60 | 41.27 | 41.77 | 41.69 | 42.44 | 42.44 | 43.08 | 43.02 | 43.02 | 43.02 | 43.02 |
| Côte d'Ivoire | 22.34 | 23.65 | 25.19 | 27.01 | 29.17 | 31.72 | 34.66 | 37.13 | 39.01 | 41.78 | 43.16 | 44.73 | 43.66 | 45.68 | 45.60 | 45.60 | 45.60 |
| Burkina Faso | 19.95 | 23.56 | 27.99 | 33.46 | 40.28 | 47.93 | 55.94 | 64.77 | 72.38 | 77.27 | 80.18 | 81.22 | 81.91 | 82.50 | 82.00 | 82.00 | 82.00 |
| Senegal | 17.91 | 18.62 | 19.39 | 20.23 | 21.14 | 22.12 | 23.68 | 25.03 | 25.71 | 25.97 | 26.51 | 26.39 | 26.50 | 26.53 | 26.53 | 26.53 | 26.53 |
| Liberia | 12.40 | 14.09 | 16.15 | 18.68 | 21.79 | 26.03 | 33.25 | 43.04 | 51.31 | 54.19 | 58.35 | 62.29 | 63.20 | 63.20 | 63.20 | 63.20 | 63.20 |
| Yemen | 10.22 | 10.84 | 11.54 | 12.32 | 13.19 | 14.17 | 15.27 | 16.19 | 18.99 | 21.81 | 21.98 | 22.46 | 23.06 | 23.65 | 23.92 | 23.90 | 23.90 |
| CAR | 6.26 | 7.44 | 8.89 | 10.65 | 12.81 | 15.46 | 18.94 | 22.41 | 26.53 | 30.42 | 32.26 | 34.41 | 35.96 | 36.05 | 36.81 | 37.08 | 37.08 |
| Nigeria | 5.30 | 6.37 | 7.71 | 9.39 | 11.51 | 14.15 | 17.13 | 20.06 | 23.34 | 26.73 | 29.10 | 30.90 | 32.27 | 33.38 | 33.54 | 33.54 | 33.54 |
| Niger | 4.36 | 3.98 | 3.67 | 3.42 | 3.24 | 3.14 | 3.13 | 3.00 | 2.91 | 3.20 | 3.13 | 3.25 | 2.79 | 2.53 | 2.50 | 2.50 | 2.50 |
| Kenya | 4.19 | 5.07 | 6.18 | 7.57 | 9.35 | 11.93 | 15.31 | 19.30 | 23.77 | 28.70 | 33.23 | 36.80 | 41.27 | 41.65 | 41.60 | 41.60 | 41.60 |
| Benin | 1.86 | 2.14 | 2.49 | 2.95 | 3.62 | 4.70 | 6.71 | 8.90 | 11.04 | 12.99 | 14.01 | 16.21 | 16.91 | 17.61 | 17.40 | 17.40 | 17.40 |
| Uganda | 1.18 | 0.95 | 0.77 | 0.64 | 0.55 | 0.52 | 0.52 | 0.71 | 1.01 | 0.95 | 0.95 | 1.02 | 1.02 | 0.90 | 0.90 | 0.90 | 0.90 |
| Tanzania | 1.15 | 1.55 | 2.13 | 2.99 | 4.30 | 5.77 | 7.87 | 10.34 | 12.78 | 15.71 | 17.72 | 18.84 | 19.32 | 19.93 | 20.31 | 20.42 | 20.42 |
| Ghana | 0.45 | 0.54 | 0.65 | 0.80 | 1.04 | 1.44 | 1.81 | 2.54 | 3.19 | 4.53 | 5.65 | 5.87 | 5.77 | 6.53 | 6.65 | 6.65 | 6.65 |
| Iraq | 0.35 | 0.58 | 0.97 | 1.66 | 2.85 | 4.97 | 5.92 | 8.36 | 9.36 | 10.44 | 9.71 | 9.53 | 9.80 | 9.80 | 9.80 | 9.80 | 9.80 |
| Togo | 0.27 | 0.34 | 0.45 | 0.60 | 0.82 | 1.15 | 1.67 | 2.87 | 4.19 | 5.58 | 6.89 | 7.69 | 8.03 | 7.95 | 7.95 | 7.95 | 7.95 |

The impact of this refined method is best illustrated by comparing it with the methods used in two previous US studies–Jones *et al.* [22] who provided the first estimate of FGM/C prevalence in the US, and Goldberg *et al.* [25] who developed the estimate currently used by US government agencies. In their study, Jones *et al.* applied the national average prevalence to the whole target population, while Goldberg *et al.* applied the national average to the target population aged over 20 and the 15–19 prevalence to those under the age of 20.

For the purposes of illustration, target population (*TP*) records extracted from the 2015–2019 American Community Survey indicated that 163,969 girls and women of Ethiopian descent were resident in the US in 2019 (see S1 Data). Of those, 39,051 were minors below the age of 15 and thus potentially still at risk of FGM/C [1]. According to the latest country-of-origin prevalence data (DHS 2016), the national-average prevalence of FGM/C in Ethiopia is 65.2% while the 15–19 prevalence was 47.1%.

The three methods were then used to estimate the scale of the impacted population. Using Jones *et al.*'s method it is estimated that 106,908 girls and women of Ethiopian descent living in the US in 2019 are impacted by FGM/C. Applying Goldberg *et al.*'s two-age-group method,

**Table 13. Comparison between methods with Jones and Goldberg based on prevalence data extracted from DHS 2016 and the third based on the mean prevalence per age group as shown in Tables 9 and 10.** (The full workings are shown in the S1 Data).

| Method | Estimated number of girls aged 0–14 at risk of FGM/C | Estimated number of women aged 15+ living with FGM/C | Total estimated population impacted by FGM/C |
|---|---|---|---|
| Jones *et al.* method | 25,461 | 81,447 | 106,908 |
| Goldberg *et al.* method | 18,393 | 79,476 | 97,869 |
| Callaghan (this paper's method) | 10,586 | 80,651 | 91,236 |

the impacted population drops to 97,869, while applying the method developed in this paper decreases the total impacted population to 91,236. Segmenting the results into two age cohorts as shown in Table 13 clearly demonstrates that previous studies overestimated the numbers of girls at risk of FGM/C.

The overestimation inherent in Jones *et al.*'s and Goldberg *et al.*'s methods is particularly noticeable in the 0–15 age-group, where this refined method results in a 58.4% (14,876) drop in the number of girls thought to be at risk when compared to calculations conducted according to Jones *et al.*'s method. On the other hand, the number of women and girls over the age of 14 living with FGM/C is similar in the three calculations.

## 4.2 Predicting future prevalence

These same calculations can be used to predict future prevalence in countries of origin, adding nuance to, and strengthening the evidence base for, analyses of progress toward the 2030 Sustainable Development Goal targets [30]. Weny *et al.* [31] point to the Gambia, Guinea Bissau, Mali and Guinea as countries making no real progress towards the eradication of FGM/C by 2030. Unicef [2] added Somalia and Senegal to that list, while classifying Guinea as making 'some progress' that would require at least a 100-fold increase to meet the 2030 target.

Applying the method described in this paper and setting the *Target Year* to 2030 suggests that nine countries will have failed to halve the risk of FGM/C to children born between 1970 and 2030, thereby adding three countries–Egypt, Sudan and Mauritania–to those identified by Weny *et al.* and Unicef. These calculations validate the method suggested herein as consistent with, but more nuanced than, those applied by other researchers.

## 4.3 The impact of migration and acculturation

Since the calculations presented in Table 12 do not take the impact of migration and acculturation into account, they should be considered maximum estimates. Two further factors would need to be considered when using those estimates in the diaspora context. The first is explicitly considered in the third variable (AI) of Yoder and Van Baelen's 'Extrapolation of FGM/C Countries' Prevalence Data' method which seeks to account for a reduction in risk post migration. Segmenting the population into three groups–those who migrated after the age of cutting, those who migrated before the age of cutting, and those born in the country of residence–suggests a differentiated impact of migration and acculturation [32]. It is clear that those who migrated after the age of cutting were at risk according to the prevalence in their country of origin while the risk to those who migrated before the age of cutting or who were born in the country of residence is impacted by the effects of migration and acculturation. The acculturation impact variable (AI) thus applies differently to each of those groups.

The second factor to consider is the potential impact of selective migration. Ortensi *et al.* suggest that those who migrate to Western countries are more likely to be urban, more educated and economically better off and propose a method by which to calculate the

differentiated prevalence for that demographic [27]. Using their method, it was estimated that a Migration Selection Factor of 0.91 applies to the Ethiopian population used to demonstrate the method above thus suggesting that those who migrated after the age of cutting where less at risk than is implied by the age-specific prevalence data.

### 4.4 Wider application

The calculations developed in this paper have been focused on FGM/C, however the same calculations could potentially be used to predict prevalence trends in other self-reported social-norms-driven data captured by MICS and DHS–such as child marriage [33] and intimate partner violence [34, 35]–which are increasingly susceptible to social desirability bias [36, 37].

### 4.5 Limitations

The extrapolation calculations presented in this paper assume a business-as-usual approach to FGM/C policy and interventions. Significant changes, either to discourage FGM/C–as was the case of Ebola-related bans in Sierra Leone [38]–or to liberalise policy–evident in efforts to overturn FGM/C legislation in The Gambia [39]–are not taken into account.

Furthermore, the extrapolation calculations presented in this paper assume a future-oriented target-population date–a date greater than or equal to the date of the most recent prevalence survey used in the calculation–and do not correctly compute prevalence estimates retrospectively. While such retrospective calculations are theoretically possible, they are not the focus of this paper.

Inherent in these calculations is the assumption that prevalence is consistent within each age group–that 9-year-olds and 5-year-olds are equally at risk for example. While this is unlikely to be true, it significantly simplifies the calculations which could in a more complex for be reformulated to take the implied trend between age groups into account, thereby recalculating the age for each single-year-group. This simplification is somewhat smoothed by the formula in step 4 which shifts single years between quinquennia but not fully accounted for.

## 5. Conclusion

This paper presents a refined method for calculating age-specific prevalence values of FGM/C, addressing the limitations of previous estimates in both the US and Europe. By disaggregating prevalence data by age, considering historical trends, and aligning prevalence values with target migrant-population data, the proposed method offers a more accurate approach for estimating the scale of FGM/C-impacted populations.

The results of applying this refined method demonstrate significant differences when compared to previous estimates. Specifically, the new method reveals a considerable overestimation in earlier studies in the US of the number of girls at risk of FGM/C.

By providing detailed calculations for the Ethiopian prevalence data and applying the proposed method to other countries of origin with nationally representative prevalence surveys, this paper offers an approach for improving the accuracy of the FGM/C prevalence variable in estimates based on Yoder and Van Baelen's 'Extrapolation of FGM/C Countries' Prevalence Data' method. Implementing this refined method can aid in better understanding the true prevalence of FGM/C in migrant populations and inform more effective interventions and policies aimed at addressing this harmful practice.

Furthermore, by predicting prevalence in 2030, this method is shown to support the findings of other researchers that specific countries will fail to meet their SDG commitments. While these calculations and predictions were focused on the practice of FGM/C, the same methodology is likely also relevant to other prevalence data.

## Supporting information

**S1 Data. Table 1: FGM/C prevalence extracted from 120 nationally representative surveys sorted by country and year of survey.**
(DOCX)

## Author Contributions

**Conceptualization:** Sean Callaghan.

**Data curation:** Sean Callaghan.

**Methodology:** Sean Callaghan.

**Writing – original draft:** Sean Callaghan.

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
