## [Decision Letter · Decision Letter 0]

29 Sep 2024

PONE-D-24-20214Calculating Age-Specific Prevalence Rates of female genital mutilation / cutting (FGM/C) for use as an input variable in Yoder and Van Baelen's ‘Extrapolation of FGM/C Countries’ Prevalence Data’ methodPLOS ONE

Dear Dr. Callaghan,

Thank you for submitting your manuscript to PLOS ONE. After careful consideration, we feel that it has merit but does not fully meet PLOS ONE’s publication criteria as it currently stands. Therefore, we invite you to submit a revised version of the manuscript that addresses the points raised during the review process.

We look forward to receiving your revised manuscript.

Kind regards,

Prof. Dr. Susanne Grylka-Baeschlin

Academic Editor

PLOS ONE

Reviewers' comments:

Reviewer's Responses to Questions

**Comments to the Author**

1. Is the manuscript technically sound, and do the data support the conclusions?

Reviewer #1: Yes

Reviewer #2: Yes

2. Has the statistical analysis been performed appropriately and rigorously? 

Reviewer #1: Yes

Reviewer #2: Yes

3. Have the authors made all data underlying the findings in their manuscript fully available?

Reviewer #1: Yes

Reviewer #2: Yes

4. Is the manuscript presented in an intelligible fashion and written in standard English?

Reviewer #1: Yes

Reviewer #2: Yes

5. Review Comments to the Author

Reviewer #1: This is a well-written article, and I enjoyed reading it. The Methods behind the results have been described with due diligence, and I find them comprehensible even for readers without an epidemiology background. The following comments focus on the structure of the article.

Major comments

1.There is not a clear distinction between the Sections: the Introduction section starts with a description of the problem, and there is a logical flow; however, this section ends abruptly with a formula, which I would expect to see in the Methods section with the necessary details to understand the formula. The Methods section introduces the data sources used and has a logical flow; however, this section is very lengthy and mixes information that could be presented in the Discussion, where the results of the present study are juxtaposed with those of previous studies. Consequently, the Discussion and Limitations sections are very small.

2.Comprehension of the results will be enhanced if the following suggested amendments are performed:

•Table 13

-Sort the countries in increasing order of the % in Age 0 to 4. This amendment will highlight the countries with lower and higher prevalence already from younger ages.

-Use the power of Excel to colour the cells with green/red shades for each country: the lower the prevalence below 50%, the darker the green, and the larger the prevalence above 50%, the darker the red. Ensure that green and red shades for the larger prevalences are not too dark, making it difficult to read the numbers. Note that the shading should be uniform for all countries; namely, the shading should not be based on each country's minimum and maximum prevalence, as it will make the Table misleading: minimum and maximum prevalence are considered 0% and 100%, respectively, in *all* countries, with shades getting 'whiter' for prevalences close to 50% from both 'directions'. Hence, countries with prevalences above 50% will have only red shades, and those with prevalences below 50% will have only green shades.

•

Table 12

-Apply the same amendments suggested for Table 13; however, sorting will refer to the Survey type *within* each country!

- Optional: First, you may sort the countries in increasing order of the average prevalence for Age 15-19, and then, sort the Survey type *within* each country

Minor comments

1.There are many single-sentence paragraphs in the Methods section that should be avoided

Reviewer #2: Thank you very much for your work, the manuscript is well written and very relevant for women health and obstetric care. I hope that this method will be used in further studies and assessments.

I think that some points need to be improved before publication, most of them are clarification of some details, please see below.

Title

In the discussion You are also suggesting that the described method can be used to assess changes over time in each country, however this does not emerge from the title.

Abstract

You may add that this method may be used to predict future prevalence of FGM/C as suggested in discussion

Section 1

• Section 1,2,3 may be included in the introduction. Section 2 and 3 may be used as subtitles in the introduction section. This will provide to the manuscript the usual structure introduction, methods, results, discussion

• You may use more recent estimates for the global female migrant population, for example https://www.migrationdataportal.org/themes/gender-and-migration

• I suggest anticipating in the text (lines 43-46) the acronyms used in the formula (Pr, TP, AI). You may also briefly describe how AI is calculated, please specify if this value change according to the country of origin.

• Please specify if TP is the absolute number or something else

Section 2

• Line 75: is “verses” a typo?

• You may add how many women participate on average to the surveys

• Could you please specify why/how you selected 27 countries among 138 countries with available data?

Section 3

• Are all the prevalences described using the formula provided in section 1? Were Goldberg using the same formula but with different Pr?

• Please describe in this section the method Jones et al cited in section 5

• Lines 114-8: from your description it seems that there are 3 estimates, one overall based on the national average, the second for the migrant population aged< 20 and the third for the migrant population aged>= 20

• Please briefly add limitations of European studies or underline why that they are not satisfactory

• Was the new method described in this manuscript previously applied by Callaghan? What is the new contribution of this paper compared to the Callaghan’s manuscript?

• Please clearly state the aim of the manuscript in lines 143-146. “further refinements” is not specific: you may move lines 148-151 here.

• Section 3 (except for the aim of the manuscript) may be moved among discussions, comparing the new approach with the existing one. Editors can provide their view on this point.

Section 4

• Please clarify if Jones (20) and Goldberg (22) use the formula provided in section 1. Please provide details on the two methods. Please note that in section 3 you describe Goldberg (22)’s method using the age cut-off of 20 while you present data for under18s in this section.

• Step 1: you may explain why the prevalence among women 50+ is not available from surveys, please clarify inclusion criteria for the survey in the previous section

• Step 2-5: as above, I suggest anticipating in the text the acronyms used in the formula (eg temp pr[10 to 14] , d, r…) you may use the italic for the acronyms

• Table 5-7: please check column names. “age 15-19” is repeated twice while “age 45-50” is missing

• Table 5 please specify the meaning of the asterisk

• Step 3-4: I’d suggest adding some details on how to interpretate q and r (q is the number of quinquennia from the survey year to the target year …). I’d suggest explaining the formula in step 4 (eg. With this formula we will calculate the prevalence in that specific age group in the target year based on data collected on …, taking into account that there will be a shift because women get older…). I’d specify in the title “aligned to the target year”

• Step 5-6: please add the reference for the methods Ortensi and Menonna and Yoder. Why is the step 2 necessary if you can extrapolate prevalence down to ages 0-4?

• Step 7: please edit as 7.3% difference (line 243). Please specify that the mean can be calculated under the assumption of fully representativeness of the survey.

• Please edit table 11b: it should be table 10b

Section 5 results

• What data were used for the first two methods in table 11 ? DHS2016 or data shown in tables 10a and 10b? please clarify if there are further adjustments to be made to data from table 10 to table 11. You may add how to calculate the Pr (formula in section 1) from table 10.

• In lines 275-6 please add absolute numbers .

• Table 1 seems a repetition of table 12. You may consider to move table 1 in the supplementary file if you maintain table 12 in the main text. Otherwise, if you may move table 12 in supplementary file as it’s the application of the steps described above (step 1)

Section 5 discussion

• note that discussion and results have the same number

• do you think that this method may find application in other fields?

Section 6

• Section 6 can be included in the discussion, no need to have a title here.

• I ‘d add the assumptions (eg there is equal prevalence in the same age group, assumption of no change in the prevalence when calculating the aligned prevalences) among limitations.

Section 7

You may add that this method may be used to predict future prevalence of FGM/C as suggested in discussion

6. PLOS authors have the option to publish the peer review history of their article (what does this mean?). If published, this will include your full peer review and any attached files.

Reviewer #1: No

Reviewer #2: No

---

## [Author Response · Author response to Decision Letter 0]

18 Oct 2024

The resubmission enclosed with this letter, Calculating Age-Specific Prevalence Rates of female genital mutilation / cutting (FGM/C) for use as an input variable in extrapolation calculations and as predictors of future prevalence in countries of origin, responds to the points raised by the reviewers in late September 2024.

Firstly, I would like to thank the reviewers for their positive feedback regarding the article. Your comments have strengthened my resubmission and widened the application of the method described in the article. I have been able to make most of the edits suggested and below respond to your specific comments:

General

• The title, abstract and conclusion of the article have been updated to reflect the widened scope of the article.

• The article has been restructured into the traditional five sections – Introduction, Method, Results, Discussion and Conclusion.

Introduction

• The global migrant data set has been updated.

• Clarity was brought my explanation of the extrapolation equation.

• Clarity and detail were added to the source data section (now labelled 1.1).

• The review of previous studies (now labelled 1.2) was reworked to better articulate the limitations of both US and European studies.

Method

• The 2019 population data used in the application of the method was moved to the discussion secion.

• Heading and descriptions in the method steps were strengthened.

• The tables headers were corrected.

• Single sentences were integrated into paragraphs.

Results

• The application to the method to the 2019 population data was moved to the discussion section.

• The long table of input prevalence data was moved to supplemental data.

• The results table showing 2024 prevalence was reordered by 0-4-year-old prevalence but not colour coded as it was felt that this would not add significantly to the understanding of the results.

Discussion

• The Discussion section was reworked and expanded.

• The application of the method to the 2019 target population using Jones, Goldberg and this method was reworked and consolidated into the new section 4.1. 

• More detailed calculations that underpin the comparison between the three methods was added as supplemental data.

• Application of the method in the wider context was explored in section 4.3.

• The limitations (4.4) were expanded.

With regards to the question from Reviewer #2 regarding the application of this method to Callaghan (2023): a version of this refined method is being used by the author in analysis for his PhD (ongoing), early results of which were published by the non-profit AHA Foundation in November 2023. This grey literature publication while based on the method described here did not articulate the methodology outlined in this paper.

---

## [Decision Letter · Decision Letter 1]

15 Dec 2024

PONE-D-24-20214R1Calculating Age-Specific Prevalence Rates of female genital mutilation/cutting (FGM/C) for use as an input variable in extrapolation calculations and as predictors of future prevalence in countries of originPLOS ONE

Dear Dr. Callaghan,

Thank you for submitting your manuscript to PLOS ONE. After careful consideration, we feel that it has merit but does not fully meet PLOS ONE’s publication criteria as it currently stands. Therefore, we invite you to submit a revised version of the manuscript that addresses the points raised during the review process.

We look forward to receiving your revised manuscript.

Kind regards,

Susanne Grylka-Baeschlin, PhD

Academic Editor

PLOS ONE

**Journal Requirements:**

Reviewers' comments:

Reviewer's Responses to Questions

**Comments to the Author**

1. If the authors have adequately addressed your comments raised in a previous round of review and you feel that this manuscript is now acceptable for publication, you may indicate that here to bypass the “Comments to the Author” section, enter your conflict of interest statement in the “Confidential to Editor” section, and submit your "Accept" recommendation.

Reviewer #1: All comments have been addressed

Reviewer #3: All comments have been addressed

2. Is the manuscript technically sound, and do the data support the conclusions?

Reviewer #1: Yes

Reviewer #3: Yes

3. Has the statistical analysis been performed appropriately and rigorously? 

Reviewer #1: Yes

Reviewer #3: Yes

4. Have the authors made all data underlying the findings in their manuscript fully available?

Reviewer #1: Yes

Reviewer #3: Yes

5. Is the manuscript presented in an intelligible fashion and written in standard English?

Reviewer #1: Yes

Reviewer #3: Yes

6. Review Comments to the Author

**Reviewer #1: **The author has addressed my comments thoroughly. The clarity and flow of the manuscript have been improved. I have no further comments.

**Reviewer #3:** The author proposes a refined method for calculating prevalence rates of Female Genital Mutilation/Cutting (FGM/C) in countries of origin and in countries with migrants affected or at risk of FGM/C. Improvements include adjusting age cohorts, extrapolating prevalence to younger age groups and considering historical trends.

The study is well written and explains in detailed steps and with examples how the methodology can be improved. All suggestions from the previous round of reviews seem to have been addressed. I have one minor suggestion:

Sociological studies have shown that not immigrant groups are not equally likely to practise FGM/C in their destination countries, owing to emigrants not being representative of the population in the country of origin (e.g. Ortensi, Farina and Mennona 2015) and owing to changing attitudes away from the country of origin (e.g. Data Collection on Female Genital Mutilation in the EU 2022). While these effects are difficult to quantify, I think they merit a mention in the discussion.

Beyond this, I was curious if there is a correlation between a reduction in FGM/C in countries with practising groups and the timing of laws passed against FGM/C in many of these countries in the last 25 years. But I appreciate that this might need to be reserved for future work.

7. PLOS authors have the option to publish the peer review history of their article (what does this mean?). If published, this will include your full peer review and any attached files.

Reviewer #1: No

Reviewer #3: No

---

## [Author Response · Author response to Decision Letter 1]

27 Dec 2024

Thank you for your encouraging response. I have separated the supplemental data from the manuscript as requested.

---

## [Decision Letter · Decision Letter 2]

7 Jan 2025

Calculating Age-Specific Prevalence Rates of female genital mutilation/cutting (FGM/C) for use as an input variable in extrapolation calculations and as predictors of future prevalence in countries of origin

PONE-D-24-20214R2

Dear Dr. Callaghan,

We’re pleased to inform you that your manuscript has been judged scientifically suitable for publication and will be formally accepted for publication once it meets all outstanding technical requirements.

Kind regards,

Susanne Grylka-Baeschlin, PhD

Academic Editor

PLOS ONE

Additional Editor Comments (optional):

Reviewers' comments:

Reviewer's Responses to Questions

**Comments to the Author**

1. If the authors have adequately addressed your comments raised in a previous round of review and you feel that this manuscript is now acceptable for publication, you may indicate that here to bypass the “Comments to the Author” section, enter your conflict of interest statement in the “Confidential to Editor” section, and submit your "Accept" recommendation.

Reviewer #1: All comments have been addressed

Reviewer #3: All comments have been addressed

2. Is the manuscript technically sound, and do the data support the conclusions?

Reviewer #1: Yes

Reviewer #3: Yes

3. Has the statistical analysis been performed appropriately and rigorously? 

Reviewer #1: Yes

Reviewer #3: Yes

4. Have the authors made all data underlying the findings in their manuscript fully available?

Reviewer #1: Yes

Reviewer #3: Yes

5. Is the manuscript presented in an intelligible fashion and written in standard English?

Reviewer #1: Yes

Reviewer #3: Yes

6. Review Comments to the Author

Reviewer #1: The author has addressed my comments thoroughly. The clarity and flow of the manuscript have been improved. I have no further comments.

Reviewer #3: (No Response)

7. PLOS authors have the option to publish the peer review history of their article (what does this mean?). If published, this will include your full peer review and any attached files.

Reviewer #1: No

Reviewer #3: No

---

## [Editor Report · Acceptance letter]

17 Jan 2025

PONE-D-24-20214R2 

PLOS ONE

Dear Dr. Callaghan, 

I'm pleased to inform you that your manuscript has been deemed suitable for publication in PLOS ONE. Congratulations! Your manuscript is now being handed over to our production team.

Kind regards, 

on behalf of

Prof. Dr. Susanne Grylka-Baeschlin 

Academic Editor

PLOS ONE